# Perceived Importance of Metrics for Agile Scrum Environments

Fernando Almeida [1,*] and Pedro Carneiro [2]

1   Faculty of Engineering, University of Porto & INESC TEC, 4200-465 Porto, Portugal
2   School of Science and Technology, Polytechnic Higher Institute of Gaya, 4400-103 Vila Nova de Gaia, Portugal; pcarneiro@ispgaya.pt
*   Correspondence: almd@fe.up.pt; Tel.: +351-225-081-400

**Abstract:** Metrics are key elements that can give us valuable information about the effectiveness of agile software development processes, particularly considering the Scrum environment. This study aims to learn about the metrics adopted to assess agile development processes and explore the impact of how the role performed by each member in Scrum contributed to increasing/reducing the perception of the importance of these metrics. The impact of years of experience in Scrum on this perception was also explored. To this end, a quantitative study was conducted with 191 Scrum professionals in companies based in Portugal. The results show that the Scrum role is not a determining factor, while individuals with more years of experience have a higher perception of the importance of metrics related to team performance. The same conclusion is observed for the business value metric of the product backlog and the percentage of test automation in the testing phase. The findings allow for extending the knowledge about Scrum project management processes and their teams, in addition to offering important insights into the implementation of metrics for software engineering companies that adopt Scrum.

**Keywords:** agile; Scrum; process measurement; metrics; software engineering; project management; information management

## 1. Introduction

Agile methodologies have gained great importance in the project management field, having been strongly influenced by Japanese philosophy. As argued by Poth et al. [1], the practices related to planning, controlling, and streamlining are actions strongly related to techniques and principles of Lean production that can be applied to any industry with the goals of reducing waste and creating value. Within the agile methodologies, we can find different methods such as Kanban, Lean, Scrum, Extreme Programming (XP), and the Rational Unified Process (RUP), among others. Data obtained by KPMG in 2019 [2] indicate that 91% of organizations consider the adoption of agile in their organizations a priority and Digital.ai [3] registered an increase from 37% in 2020 to 86% in 2021 in the number of agile adoptions in software teams, with Scrum standing out as the most popular framework, followed by Kanban and Lean.

The agile manifesto emerged in 2001 when a group of 17 representatives from various software development practices and methodologies met to discuss the need for lighter and faster alternatives to the existing traditional methodologies. From this meeting, the Agile Alliance presented the Manifesto for Agile Software Development to elucidate the approach known today as agile development. The values of the agile methodology are based on four pillars [4]: (i) individuals and interactions over processes and tools; (ii) working software over comprehensive documentation; (iii) customer collaboration over contract negotiation; and (iv) responding to change over following a plan. Agile methods were designed to use a minimum of documentation, helping in the flexibility and responsiveness to change, that is, in this methodology, flexibility and adaptability are much more important than planning, unlike the traditional methodology [5–10].

Scrum was conceived by Jeff Sutherland and Ken Schwaber in 1993 with the intention of being a faster, more effective, and more reliable way to develop software for the technology industry [11]. This method emerged motivated by the traditional method, called waterfall, being too slow and often resulting in a product not desired by the customer and more expensive [12,13]. Alternatively, agile methodologies present an incremental and iterative process whose objective is to identify the priority tasks in each phase and effectively manage time with efficient teams [14–18]. Therefore, agile methodologies came to face the difficulties that occurred during project management.

The processes inherent to the several phases of process management must be effective and efficient. According to Flores-Garcia et al. [19], the advances in technology that have occurred in the last decades have provided business managers with a large volume of tools to help them make decisions. This new business scenario has forced development companies to constantly seek technologies and methods that allow them to guarantee the quality of the products offered so that they do not lose competitiveness in the market where they operate. In this context, the use of metrics that can define the performance of projects, as well as the products resulting from them, has taken on an increasingly important role in the industry and, consequently, in academia, which is preparing to meet the challenges posed by these organizations.

The most well-known and commercially successful software companies such as Google or IBM have adopted metrics to evaluate the success of their project management and campaigns [20,21]. Indeed, it is not only in the waterfall development model that it is necessary to have metrics to evaluate the performance of your processes and teams. In addition, in agile development, it is necessary to measure the effectiveness and efficiency of the processes using metrics. Planning and monitoring are necessary for projects developed in Scrum [22]. Previous works developed by Almeida & Carneiro [23], Kurnia et al. [24], and López et al. [25] were important in synthesizing these metrics considering the whole Scrum development cycle and its ceremonies. However, none of these studies provide a measure of the relative importance of these metrics considering the various agile Scrum roles (i.e., Product Owner, Scrum Master, and development team). Understanding the importance of these metrics while considering the specific role of each Scrum role is important to increase team cohesion and the quality of the work produced. It is also a way to gain practical insight into the role of each Scrum role in teams and establish policies to promote increased effectiveness and efficiency of project management in Scrum. Therefore, we have developed a quantitative study based on the perception of the importance of metrics considering several profiles of actors in the Scrum process. The rest of this manuscript is organized as follows: in the first phase, a literature review is performed on metrics that can be found in a Scrum environment. After that, the several methodological phases of the study are presented. This is followed by the presentation and discussion of the results, considering their contribution to the increase of knowledge in the field. Finally, the main conclusions are listed, considering their theoretical and practical contributions. It is also in this last section that the main limitations of the study are exposed and suggestions for future work are presented.

## 2. Background

Velimirovic et al. [26] note that to monitor the progress of projects and promote the necessary improvements during their execution, the use of performance indicators is necessary. Therefore, before starting project management, the first step to ensure success is to define the metrics for follow-up. Having performance indicators for each stage of project implementation is essential to optimize the results and guide the team's path. In the software industry, metrics are used for several reasons, such as project planning and estimation, project management and monitoring, understanding quality and business goals, and improving communication, processes, and software development tools [27].

The first paper identifying metrics for the Scrum environment was developed in 2015 by Kupiainen et al. [28]. This study sought to provide identifications collected from scientific

papers and metrics used in the industry. Despite this dual aspect, the methodology used did not involve the collection of primary metrics but only a survey of secondary sources through the development of a systematic literature review. The rationale for and effects of the use of metrics in areas such as sprint planning, software quality measurement, impediment resolution, and team management were identified. The results of this study allowed us to conclude that the most important metrics are related to customer satisfaction and backlog progress status monitoring, considering the product specifications and the work developed in each sprint.

Other studies have been undertaken. Kurnia et al. [24] collected 34 metrics related to Scrum development. The metrics included the entire Scrum development cycle, such as sprint planning, daily meetings, and retrospective sessions. The findings contributed to the identification of the most common metrics in the literature and identified new metrics related to the value delivered to the customer and the rate of development throughout the sprint.

In Almeida & Carneiro [23], a review of Scrum metrics was performed considering a primarily quantitative study with software engineers. The study involves 137 Scrum engineers and concluded that "delivered business value" and "sprint goal success" were the most relevant metrics.

In López et al. [25], a total of 61 studies from the last two decades were reviewed to explore how quality is measured in Scrum. Two important conclusions were drawn from this study. First, despite a large body of knowledge and standards, there is no consensus regarding the measurement of software quality. Another conclusion is that there is a very diverse set of metrics for this purpose, with the top three being related to performance, reliability, and maintainability.

In Kayes et al. [29], a different perspective was used, with the goal of proposing a metric for measuring the quality of the testing process in Scrum. Therefore, instead of looking at the whole Scrum cycle, attention was focused on a specific action. The Product Backlog Rating (PBR) provides a complete picture of the testing procedure of a product throughout its development cycle. The PBR takes into consideration the complexity of the features to be produced in a sprint, evaluates the test ratings, and provides a numerical score of the testing process. A similar line of research is the work conducted by Erdogan et al. [30], which looked at the value of metrics in the process of analyzing sprint retrospectives. This study found metrics for the inspection process, improving team estimates and increasing team productivity. This study further found that the metrics of "actual quality activity effort rate" and "subcomponent defect density" helped improve product quality.

The metrics collected as part of this study allowed us to synthesize the metrics and their definition, as shown in Table 1. Metrics related to effort estimation can be used to prioritize features to be developed or to prioritize activities based on relative value and effort, and velocity can be used to improve effort estimates for the next iteration, which will help the team to verify that the planned scale has been completed. Metrics related to defect identification can be used to inspect the defects in the backlog, which will allow the sharing of this information among the team members. In addition, in the same vein, we found metrics that measure defects that appear during a sprint. Finally, there are metrics, such as return on investment, that measure the delivery of software and that can be used to understand the relationship between the result and the investment in software. These metrics apply to the Scrum software development process and do not specifically explore their importance considering the three Scrum roles (i.e., Product Owner, Scrum Master, and development team), which prompts us to establish the first research hypothesis.

**Table 1.** Overview of Scrum metrics.

| Activity | Metric | Definition |
| --- | --- | --- |
| Daily scrum | Number of tasks | Total number of tasks in the sprint backlog. |
| | Number of tasks in progress | Number of tasks "in progress" in the sprint backlog. |
| | Number of concluded tasks | Number of tasks completed in the sprint backlog. |
| | Estimated hours for a task | Time needed in hours to complete a task. |
| | Remaining hours for a task | Remaining time in hours to complete a task. |
| | Number of impediments | Number of impediments, obstacles, or issues that hinder the progress of a Scrum team during the implementation of a sprint. |
| | Workload distribution | Measure of how much work is assigned to each development member for the current sprint. |
| Product backlog | Number of user stories | Total number of user stories in the product backlog. |
| | Number of added user stories | Number of new user stories added to the product backlog. |
| | Number of deleted user stories | Number of user stores removed from the product backlog. |
| | Business value | Importance of a user story considering the product owner's vision. It should reflect the value generated for the organization in terms of revenue, customer satisfaction, market share, competitive advantage, or any other relevant business objective. |
| Sprint backlog | Number of user stories | Number of user stories in the sprint backlog. |
| | Number of tasks | Number of tasks in the sprint backlog. |
| | Hours spent to implement a task | Hours spent in a day to implement a given task. |
| | Hours remaining to finish a sprint | Time in hours remaining to finish the current sprint. |
| | Sprint burndown | Graphic representation of the rate at which work is completed and how much work remains to be performed in a sprint. |
| Sprint planning meeting | Sprint length | Duration of a given sprint. |
| | Size of team | Number of developers in the development team. |
| | Team members' engagement | Level of engagement of the team member in their work and workplace. |
| Sprint retrospective | Number of tasks in a sprint | Number of tasks assigned to a sprint. |
| | Number of tasks completed in a sprint | Number of tasks completed in a sprint. |
| | Number of user stories completed in a sprint | Number of user stories implemented during the sprint. |
| Sprint review | Number of accepted user stories | Number of user stories accepted by the customer during the sprint review. |
| | Number of rejected user stories | Number of user stories rejected by the customer during the sprint review. |
| | Accuracy of estimation | Percentage of correctness of the estimated implementation time of the user stories compared to their actual implementation. |
| Team performance | Focus factor | The speed of implementation to be divided by the internal capacity of the team. |
| | Targeted value increase | Team's speed in the current sprint divided by its initial speed. |
| | Team member turnover | Indicates the turnover of team members considering a full development cycle. |
| | Team satisfaction | Degree of satisfaction of the team with the Scrum environment and adopted methodologies. |
| | Velocity | Amount of work a development team can do during a sprint. It can be calculated by considering the story points divided by actual hours or the estimated hours divided by actual hours. |
| | Work capacity | The total time the team is available for work during a sprint. It is usually measured in hours. |
| Tests | Acceptance tests per user story | Number of acceptance tests per user story. |
| | Defects count per user story | Total number of defects per user story |
| | Defects density | Number of defects found divided by the size of the considered module/software. |
| | Functional tests per user story | Number of functional tests per user story. |
| | Tests automation percentage | Tests automation percentage considering automatic tests and manual tests. |
| | Unit tests per user story | Number of unit tests per user story. |

**H1:** *There are significant differences in the perceived importance of Scrum metrics according to the Scrum role.*

Scrum governance is a challenging task because it cannot focus only on software development processes and must involve multiple domains and include interdisciplinarity members. Empirical studies developed by [31–34] show that organizations implementing Scrum must concentrate their efforts on process improvement in a controlled and limited number of areas to face the high complexity of the continuous improvement processes and the strong interconnection between them. This requires keeping track of the metrics of the Scrum activities. As reported by Kadenic et al. [35], the experience of a Scrum professional in understanding the Scrum work processes is an important element for the success of the implementation and diffusion of Scrum in the organization. In this sense, this study also aimed to understand if years of experience in Scrum are important to understand the relative importance of Scrum metrics. Two further research hypotheses were established:

**H2:** *The number of years of Scrum experience is a differentiating element in the perception of the importance of Scrum metrics.*

**H3:** *Years of experience in their specific Scrum role is a differentiating element in the perceived importance of Scrum metrics.*

## 3. Materials and Methods

Figure 1 schematizes the three phases of the methodological process. In the first phase, a literature review is conducted on the main activities associated with Scrum development and the metrics that we can find to measure the effectiveness of these activities. It is also in this phase that the research hypotheses are established according to the existing gaps in the area. In the exploration stage, the questionnaire is built and distributed among Portuguese software engineering professionals who apply the Scrum methodology to manage and develop software projects. Finally, the analysis stage is responsible for the statistical analysis of the data. This is followed by the discussion of the results, which allows us to explore the main innovative contributions of this work and its importance to the knowledge about Scrum methodology. Finally, the main conclusions of the study are drawn.

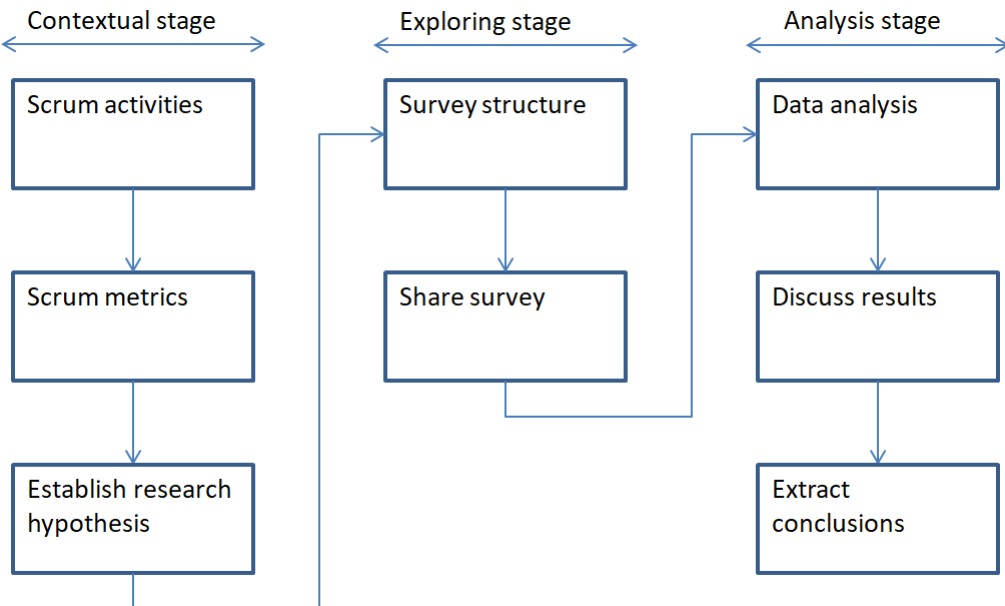

**Figure 1.** Phases of the adopted methodology.

A questionnaire was distributed through the partner network of the Portuguese Chamber of Commerce and Industry in the field of software engineering to collect the perception of the relative importance of metrics in each Scrum activity. The questionnaire

was sent by email and was also shared on the LinkedIn social network. The questionnaire was only completed by companies adopting the Scrum methodology. It was available online between 6 February 2023 and 30 March 2023. A total of 191 valid responses were received. Only partially completed questionnaires were not included in the data analysis process.

The questionnaire data were statistically explored using SPSS v.21, The Cronbach's Alpha, Composite Reliability (CR), and Average Variance Extracted (AVE) of each construct were calculated to determine its internal consistency. According to Shrestha [36], the first two coefficients should be higher than 0.7, while it is recommended that the AVE is higher than 0.6. It is seen from Table 2 that all constructs have a Cronbach's Alpha and CR greater than 7. Additionally, the AVE is greater than 0.6 in all constructs.

**Table 2.** Reliability analysis of the constructs.

| Construct | Cronbach's Alpha | CR | AVE |
|---|---|---|---|
| Control variables | 0.722 | 0.836 | 0.641 |
| Daily Scrum | 0.848 | 0.893 | 0.636 |
| Product backlog | 0.828 | 0.871 | 0.670 |
| Sprint backlog | 0.821 | 0.874 | 0.661 |
| Sprint planning meeting | 0.737 | 0.849 | 0.629 |
| Sprint retrospective | 0.788 | 0.862 | 0.688 |
| Sprint review | 0.713 | 0.854 | 0.670 |
| Team performance | 0.866 | 0.910 | 0.659 |
| Tests | 0.833 | 0.885 | 0.673 |

An ordinal scale (i.e., very low, below average, average, above average, very high) was considered and then converted for statistical analysis purposes to a Likert scale from 1 to 5.

To enable answering the previously formulated research hypotheses, it was necessary to collect information on three control variables: (i) role in the Scrum team; (ii) number of years of experience in Scrum; and (iii) number of years of experience in their current Scrum role. Table 3 presents the distribution of the sample considering the profile of the respondents. The majority of respondents play a role on the development team, which is almost double the number of respondents who play the role of Product Owner, which is consistent with the number of individuals expected on a Scrum team where the majority of the team is composed of development team members. It is also noted that more than 43% of the individuals have more than 5 years of experience in Scrum. Consistent with this information is the number of years in the same Scrum role. Nevertheless, there is a greater homogeneity of the sample in this third question, indicating that individuals may assume several Scrum roles throughout their professional careers.

**Table 3.** Sample characteristics.

| Variable | Absolute Frequency | Relative Frequency |
|---|---|---|
| What is your role? | | |
| Product Owner | 47 | 0.246 |
| Scrum Master | 66 | 0.346 |
| Development team | 78 | 0.408 |
| How many years of experience in Scrum? | | |
| Less than 1 year | 23 | 0.120 |
| Between 1 and 2 years | 27 | 0.141 |
| Between 3 and 4 years | 58 | 0.304 |
| More than 5 years | 83 | 0.435 |
| How many years of experience in your current Scrum role? | | |
| Less than 1 year | 26 | 0.136 |
| Between 1 and 2 years | 39 | 0.204 |
| Between 3 and 4 years | 61 | 0.319 |
| More than 5 years | 65 | 0.340 |

## 4. Results

The results in Table 4 provide an understanding of the perceived importance of each Scrum metric. It is calculated as the mean, mode, and standard deviation. The mode represents the most frequent response given by the respondents. Only two metrics have a perceived importance higher than 4.2 (i.e., business value and number of impediments). Furthermore, in these two situations, the mode equals 5, which indicates that most of the respondents indicated the maximum importance of these two metrics. Other metrics (i.e., number of user stories, number of concluded tasks, number of tasks completed in a sprint, number of user stories completed in a sprint, accuracy of estimation, and velocity) also have a mode of 5, although their average is lower than 4.2. The standard deviation analysis allows us to see the homogeneity of the respondents' behavior. The largest dispersion of answers is registered for the metric "number of deleted user stories", while the metric "functional tests per user story" presents a standard deviation around half of the highest value, indicating a greater homogeneity of answers for this metric.

**Table 4.** Statistical analysis of the importance of Scrum metrics.

| Activity | Metric | Median | Mode |
|---|---|---|---|
| Product backlog | Number of user stories | 4 | 5 |
| | Number of added user stories | 4 | 4 |
| | Number of deleted user stories | 3 | 4 |
| | Business value | 5 | 5 |
| Sprint planning meeting | Sprint length | 4 | 4 |
| | Size of team | 4 | 4 |
| | Team members' engagement | 4 | 4 |
| Sprint backlog | Number of user stories | 4 | 5 |
| | Number of tasks | 4 | 4 |
| | Hours spent to implement a task | 4 | 4 |
| | Hours remaining to finish a sprint | 4 | 4 |
| | Sprint burndown | 4 | 3 |
| Daily Scrum | Number of tasks | 3 | 3 |
| | Number of tasks in progress | 4 | 3 |
| | Number of concluded tasks | 4 | 5 |
| | Estimated hours for a task | 3 | 3 |
| | Remaining hours for a task | 3 | 3 |
| | Number of impediments | 5 | 5 |
| | Workload distribution | 3 | 3 |
| Sprint review | Number of accepted user stories | 4 | 3 |
| | Number of rejected user stories | 3 | 3 |
| Sprint retrospective | Number of tasks in a sprint | 3 | 3 |
| | Number of tasks completed in a sprint | 4 | 5 |
| | Number of user stories completed in a sprint | 4 | 5 |
| Team performance | Accuracy of estimation | 4 | 5 |
| | Focus factor | 4 | 4 |
| | Targeted value increase | 4 | 4 |
| | Team member turnover | 4 | 3 |
| | Team satisfaction | 4 | 4 |
| | Velocity | 4 | 5 |
| | Work capacity | 4 | 4 |
| Tests | Acceptance tests per user story | 4 | 4 |
| | Defects count per user story | 4 | 4 |
| | Defects density | 4 | 4 |
| | Functional tests per user story | 4 | 4 |
| | Test automation percentage | 4 | 4 |
| | Unit tests per user story | 4 | 4 |

Figure 2 complements the previous analysis by considering the relative importance of the metrics for each activity. The activities were ranked according to their position in

the Scrum work methodology. It was concluded that metrics related to team performance evaluation are the most important, followed by those related to testing. It is noteworthy that metrics related to the final phases of the Scrum process, which enable a more comprehensive perception of the organizations' strategy in managing Scrum processes and their teams, are more relevant than metrics in the early phases of the process and related to operational Scrum processes such as the daily Scrum, sprint backlog, or product backlog.

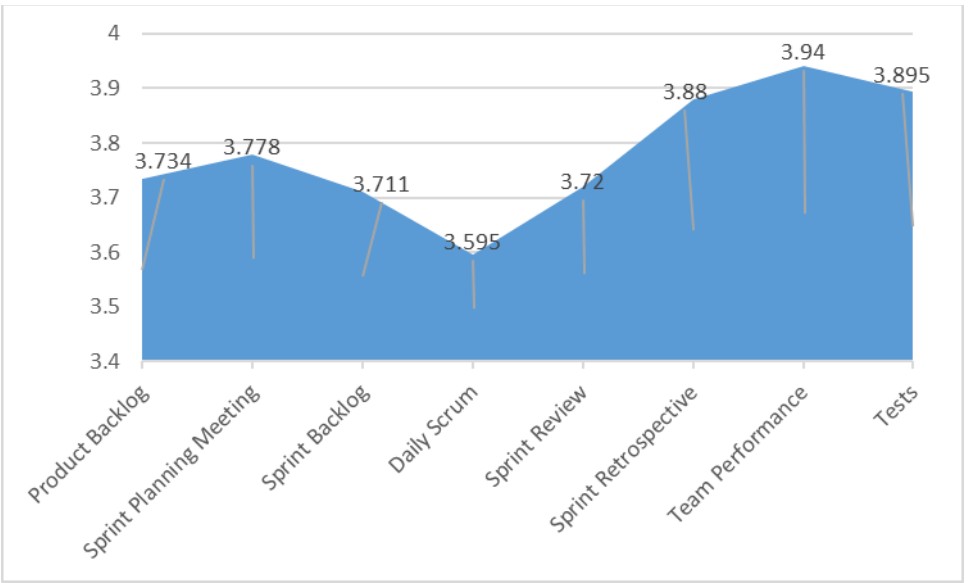

**Figure 2.** Perceived importance of metrics by activity.

Table 5 performs a statistical analysis of variance between the groups of collected responses. Three experimental parameters have been considered: (i) role in the Scrum team; (ii) years of experience (YE) in Scrum; and (iii) years of experience in the same Scrum role (YE-SR). An F-test was performed to determine the probability that the variances of two different samples are equal. The F-test uses a statistic known as the F-statistic to test statistical hypotheses about the variances of the distributions from which the samples were drawn. A significance level of 5% was considered. The findings indicate that:

- The role of the individual is not a differentiating factor in any metric. The significance of the test for the "Role" column is higher than 0.05. Therefore, H1 is rejected;
- YE is a determining factor for all metrics within the "Team Performance" activity, and even significant for the "business value" metric of the "Product Backlog" activity and the "test automation percentage" metric of the "Tests" activity. In all these situations, the significance of the test is lower than 0.05. Accordingly, H2 is accepted;
- YE-SR is also a determinant for all metrics previously considered in H2. The significance of the test is lower than 0.05 for those metrics. Therefore, H3 is also accepted. Because the significance of the metrics in H3 is completely coincident with H2, then knowledge of years of experience in the same Scrum role is not a differentiating element for understanding the importance of metrics in a Scrum environment.

**Table 5.** Statistical analysis of variance between groups.

| Activity | Metric | Role | | YE | | YE-SR | |
|---|---|---|---|---|---|---|---|
| | | F Value | Sig. | F Value | Sig. | F Value | Sig. |
| Product backlog | Number of user stories | 1.677 | 0.203 | 2.003 | 0.142 | 1.915 | 0.152 |
| | Number of added user stories | 1.428 | 0.239 | 1.735 | 0.195 | 1.679 | 0.209 |
| | Number of deleted user stories | 2.581 | 0.102 | 3.118 | 0.079 | 3.300 | 0.071 |
| | Business value | 2.784 | 0.081 | 10.916 | $<1.10^{-3}$ | 9.875 | $<1.10^{-3}$ |

**Table 5.** *Cont.*

| Activity | Metric | Role | | YE | | YE-SR | |
|---|---|---|---|---|---|---|---|
| | | F Value | Sig. | F Value | Sig. | F Value | Sig. |
| Sprint planning meeting | Sprint length | 1.566 | 0.225 | 2.012 | 0.133 | 2.455 | 0.083 |
| | Size of team | 1.311 | 0.288 | 1.515 | 0.229 | 1.890 | 0.160 |
| | Team members' engagement | 1.561 | 0.227 | 1.684 | 0.205 | 1.788 | 0.194 |
| Sprint backlog | Number of user stories | 1.733 | 0.196 | 2.056 | 0.133 | 2.158 | 0.129 |
| | Number of tasks | 1.688 | 0.200 | 2.122 | 0.124 | 2.237 | 0.115 |
| | Hours spent to implement a task | 1.820 | 0.168 | 1.900 | 0.157 | 2.245 | 0.113 |
| | Hours remaining to finish a sprint | 1.711 | 0.198 | 1.967 | 0.151 | 2.156 | 0.130 |
| | Sprint burndown | 1.555 | 0.229 | 1.890 | 0.162 | 1.908 | 0.155 |
| Daily Scrum | Number of tasks | 1.232 | 0.301 | 1.505 | 0.237 | 1.670 | 0.220 |
| | Number of tasks in progress | 1.455 | 0.232 | 1.788 | 0.188 | 1.712 | 0.199 |
| | Number of concluded tasks | 1.347 | 0.294 | 1.711 | 0.197 | 1.600 | 0.228 |
| | Estimated hours for a task | 1.670 | 0.203 | 1.990 | 0.153 | 2.103 | 0.137 |
| | Remaining hours for a task | 1.870 | 0.163 | 2.246 | 0.110 | 2.056 | 0.150 |
| | Number of impediments | 1.824 | 0.167 | 2.198 | 0.118 | 2.256 | 0.110 |
| | Workload distribution | 1.569 | 0.226 | 1.756 | 0.193 | 1.790 | 0.191 |
| Sprint review | Number of accepted user stories | 1.522 | 0.231 | 1.678 | 0.208 | 1.890 | 0.161 |
| | Number of rejected user stories | 1.967 | 0.150 | 2.289 | 0.096 | 2.099 | 0.139 |
| Sprint retrospective | Number of tasks in a sprint | 2.455 | 0.113 | 2.565 | 0.088 | 2.450 | 0.086 |
| | Number of tasks completed in a sprint | 2.311 | 0.130 | 2.812 | 0.083 | 2.491 | 0.084 |
| | Number of user stories completed in a sprint | 1.915 | 0.153 | 2.450 | 0.101 | 2.255 | 0.110 |
| Team performance | Accuracy of estimation | 1.240 | 0.297 | 5.784 | $<1.10^{-3}$ | 7.122 | $<1.10^{-3}$ |
| | Focus factor | 1.233 | 0.301 | 8.120 | $<1.10^{-3}$ | 8.770 | $<1.10^{-3}$ |
| | Targeted value increase | 1.499 | 0.226 | 7.665 | $<1.10^{-3}$ | 8.233 | $<1.10^{-3}$ |
| | Team member turnover | 1.367 | 0.291 | 9.125 | $<1.10^{-3}$ | 7.990 | $<1.10^{-3}$ |
| | Team satisfaction | 1.299 | 0.285 | 7.900 | $<1.10^{-3}$ | 7.458 | $<1.10^{-3}$ |
| | Velocity | 1.317 | 0.287 | 4.752 | 0.006 | 5.341 | 0.002 |
| | Work capacity | 1.567 | 0.228 | 5.890 | $<1.10^{-3}$ | 7.111 | $<1.10^{-3}$ |
| Tests | Acceptance tests per user story | 1.671 | 0.204 | 1.878 | 0.173 | 2.156 | 0.133 |
| | Defects count per user story | 1.502 | 0.238 | 1.923 | 0.162 | 2.178 | 0.130 |
| | Defects density | 1.788 | 0.177 | 1.998 | 0.158 | 2.091 | 0.141 |
| | Functional tests per user story | 1.245 | 0.295 | 1.652 | 0.213 | 1.890 | 0.161 |
| | Test automation percentage | 1.348 | 0.297 | 8.239 | $<1.10^{-3}$ | 7.799 | $<1.10^{-3}$ |
| | Unit tests per user story | 1.290 | 0.289 | 1.566 | 0.225 | 1.670 | 0.201 |

## 5. Discussion

### 5.1. The Importance of Metrics

The organizations' need for agility has led them to explore new ways of managing projects based on agile methodologies. Scrum has been widely disseminated and explored by IT companies in the construction of their technology solutions. However, the search for agility should not compromise the efforts of evaluation and continuous improvement of software engineering processes. The collection of metrics assumes itself as a fundamental strategy for organizations to improve their work processes [37–41]. The findings of this study confirmed the importance that metrics collection assumes for Scrum teams. However, metrics do not assume the same importance in all phases of Scrum. Metrics related to the daily execution phases of Scrum (e.g., "daily Scrum") are perceived as less important than those related to planning (e.g., "sprint planning meeting") or retrospective (e.g., "sprint retrospective") phases. Furthermore, metrics related to team performance and the testing process are those that received the highest importance from Scrum professionals.

Team performance is mainly evaluated by considering the metrics of velocity, targeted value increase, and work capacity. Velocity is a widely known and used metric in Scrum that measures the work rate [42]. Using this metric in isolation can prove detrimental as Doshi reports [43]: "If two teams have similar skillset, shouldn't their velocities be

similar?", "Team A's Velocity is 2 times that of Team B's—shouldn't Team A work on the remaining Product Backlog Items for faster delivery?" The answer to this question lies in the differences in the starting point between both teams and the estimates made for each user story. Thus, speed comparisons between the two teams can be a metric with negative effects and make the teams uncomfortable [43]. Another equally relevant metric is the "targeted value increase" that results from an analysis of the speed increase along the process. In Downey & Sutherland [44], this metric is defined as "Current Sprint's Velocity ÷ Original Velocity". Unlike velocity, which measures the teams' current performance, this metric allows for exploring process improvements. Finally, the work capacity measures the total time the team is available to work during a sprint [45]. In [46], it is suggested that team performance metrics should be combined in a team performance histogram, in which the most popular metric, such as velocity, can be combined with others, such as predicted and actual capacity. By incorporating additional metrics, teams can gain deeper insights into their performance, make more informed decisions, and improve their overall efficiency and effectiveness [47,48]. This approach is especially useful to complement the calculation of metrics such as "Velocity", which is strongly unstable, inconsistent across teams, and context dependent. For example, by combining velocity with other metrics such as backlog size, cycle time, or lead time, it becomes easier to predict future delivery dates or release milestones. This helps stakeholders and product owners to plan and more accurately set expectations. Another suggestion is to combine velocity with metrics like team capacity and individual workload to help identify if the team is overburdened or underutilized. By analyzing the relationship between velocity and capacity, teams can make better decisions about how much work to take on in each sprint and effectively allocate resources.

The testing phase is another area highlighted as offering the best conditions to be measured. The percentage of test automation is the metric that gathered the most interest from the respondents. In general, test automation consists of the use of tools to gain control over the execution of tests, allowing an increase in the coverage of software testing [49–52]. Considering the major goals of test automation, the ease of control over test execution and the reduction of resources are major attractions when one wants to implement this approach in software engineering. However, these are not the only advantages of this model. As reported in [53], automation is also able to provide consistency and reliability in test repetition, because validations are always performed in the same way; easy access to information related to test execution, because the use of automation tools makes it easier to extract data such as test progress, error rates, and performance; and, mainly, reduction in repetitive manual work, a task that over time can lose reliability, causing a decrease in software quality.

*5.2. The Impact of Control Variables*

The results of this study provided an understanding of the importance of these metrics in light of the role and years of experience of a Scrum professional. The role played by the individual in the Scrum team has no impact on the perceived importance of these metrics because the significance of the test is higher than 0.05. Scrum stands out as an inflexible methodology in which the roles of each individual are less important than working toward the success of the deliverables. Harmony emerges as a fundamental factor for successful delivery. In the work developed by Aldrich et al. [54], the role that the Scrum Master has in promoting harmony between the developers and the Product Owner is highlighted. Visibility, inspection, and adaptation are three core elements of the Scrum philosophy [55,56]. They contribute to making all professionals feel part of the whole project and not overly focused on a specific task. Consequently, this makes all Scrum practitioners, regardless of their role, recognize the importance of metrics in each activity in a similar way.

On the contrary, years of experience in Scrum is a factor that impacts the perceived importance of Scrum metrics. However, this impact is only reflected in the metrics related to team performance and in two specific metrics of the product backlog (i.e., business

value) and tests (i.e., test automation percentage) that present a significance of the test lower than 0.05. Benchmarking team performance is a process associated with continuous improvement, as reported in [57]. The perceived importance of this phenomenon is not equally recognized by all individuals. Individuals with fewer years of experience in Scrum tend to give less importance to measuring team performance and focus on the operational factors of technical implementation of Scrum. In the same vein, the business value is a metric that individuals with less experience in Scrum have less difficulty recognizing, not least because Product Owner positions tend to be held by individuals with more years of experience. The percentage of test automation is the last metric that showed significant differences. The gains from implementing test automation strategies are not limited to development processes but impact other areas such as version management, teams, and execution environments [58]. All these dimensions are more recognized by professionals with more than years of experience. Finally, the impact of years of experience in the same Scrum role was also assessed and it was concluded that its effects are negligible. Effectively, the dynamics of adaptability and progression within a Scrum environment make the diversity of performance of a Scrum professional broader and not restricted to a specific role. In this sense, the importance of metrics is not affected by this factor.

## 6. Conclusions

### 6.1. Theoretical Contributions and Practical Implications

This study offers relevant theoretical contributions by extending the literature on the adoption of metrics for evaluating agile development processes in Scrum by exploring the impact of the Scrum professional's role and experience of involvement in Scrum teams. The study concluded that the role played by the individual in the Scrum environment is not a relevant factor in the perception of the importance of metrics, while years of experience is a relevant factor in the perception of the importance of metrics related to team performance analysis activities. It was also found that the business value metric of the product backlog and the test automation percentage of the testing phase are metrics more valued by individuals with more years of experience in Scrum. It was also concluded that years of experience in the same Scrum role is not relevant in perceiving these differences.

In the practical dimension, the results of this study can be adopted by organizations that are adopting Scrum in their software development methodology or that intend to migrate in the near future. Due to the high complexity of collecting and processing metrics, it is important that organizations can focus on the metrics that offer the greatest visibility into Scrum processes and that enable teams to continuously improve their software engineering processes.

### 6.2. Limitations and Future Research Directions

This study presents some limitations that it is relevant to address. First, the study focuses specifically on the Scrum methodology. In future work, it is suggested that a similar approach be applied to other agile methodologies such as XP, RUP, Feature Driven Development (FDD), and Lean Software Development (LSD), among others. It becomes relevant specifically to explore the paces that can be found in each of these frameworks, which are different from what happens in the Scrum environment. Equally relevant as future work would be to consider the adoption of Scrum in large-scale environments, in which the complexity of process management is greater. In this sense, metrics can play an even more relevant role to have greater visibility over the processes. Another limitation is the difficulty of including all metrics in the study that include all the recommendations addressed in the literature such as story points, function points, and COSMIC function points. Furthermore, this study only measures an individual's prior experience with Scrum, ignoring that many individuals, before adopting agile methodologies, have experience in waterfall development environments. Consequently, it becomes relevant to explore whether this prior experience in traditional waterfall development and project management contributes positively or negatively to the perception of the importance of metrics in Scrum.

The difficulty in formally defining Scrum metrics is another limitation. In this sense, the respondents' perceptions of the importance of metrics are strongly related to the way these metrics were implemented in their organizations and not to the formal definition associated with each one of them. This study did not obtain information about the size and organizational structure of the companies. Scrum implementation models can be very diverse, given the size of each company and the constitution of their teams, which are increasingly geographically distributed. Exploring the impact of these two facts is also a relevant suggestion for future work. Finally, this study collects the perceptions of the various actors in the Scrum process and, consequently, does not collect information about the impact of metrics on the success of software engineering processes. In future work, it would be relevant to explore the impact of collecting each metric on software development processes.

**Author Contributions:** Conceptualization, F.A. and P.C.; investigation, F.A.; writing—original draft, F.A.; supervision, P.C. All authors have read and agreed to the published version of the manuscript.

**Funding:** This research received no external funding.

**Institutional Review Board Statement:** Not applicable.

**Informed Consent Statement:** Not applicable.

**Data Availability Statement:** Data are available on request from the authors.

**Conflicts of Interest:** The authors declare no conflict of interest.

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
