# Peer review of "Perceived Importance of Metrics for Agile Scrum Environments"

_information, doi:10.3390/info14060327_

Round 1

Reviewer 1 Report

This paper is well written. The contributions mad win this paper are helpful and representative. The paper contains information that provides some degree of archival values. Repeatability is a bit low since the method is sound but the experimental parameters are summarized only.

Author Response

We appreciate the review suggestions and comments received by the reviewer. These elements are key to improving the final quality of the manuscript. Below we respond to each issue raised.

Review #1

This paper is well written. The contributions made in this paper are helpful and representative. The paper contains information that provides some degree of archival values. Repeatability is a bit low since the method is sound but the experimental parameters are summarized only.

Author’s response: Thanks for your very positive evaluation regarding the relevance and quality of this paper. We would like to clarify that three experimental parameters have been explored in this study: (i) role in the Scrum team; (ii) years of experience in Scrum; and (iii) years of experience in the same Scrum role. We have clarified the use of these three experimental parameters.

Reviewer 2 Report

Title: The current title uses the term ‘Relevance’ (used 9 times in this paper’ while the Abstract refers to ‘Importance’ (used 25 times in this paper): therefore, the title should be modified to ‘importance’. Furthermore, this study input is the ‘opinion’ of the SCRUM practitioners’: therefore the title should be amended to ‘Perceived Importance’ instead of ‘Relevance’.

Also verify the full text to ensure that the right terms (either relevance and importance) and used in a consistent manner throughout the text.

Table 1: I was surprised to find out that  this table does not include the metrics ‘Story Points’ widely used across the industry (and mentioned often as well in the literature), nor Function Points and COSMIC Functions Points, used by a number of practitioners and sometimes mentioned in the literature. 

Table 1: I was puzzled by the definitions of ‘velocity’ and ‘work capacity’ which use ‘amount of work’ and ‘sum of all work’ where ‘work’ has not been defined-measured anywhere (most previous definitions refer to ‘tasks’, ‘user stories’, etc.

Table 4: The authors did not explicitly described how the practitioners provided their opinions of the ‘relevance-importance’ of the ‘metrics’ listed: it appears however that they have used a range from 0 to 5, in increment of 1 (of course, this is my ‘guess’, and it may be something else…): to be noted that 0 to 5 in this context is an ‘ordinal scale’: each one being ‘more important’ than the previous ones, but these are ‘ordered labels’ and not ‘numerical values on the ratio-scale type where for instance 4 = 4 times greater than 1).  Therefore, if this is the range used by the authors in their questionnaire, than:

-           it is not allowed to calculate an ‘average’ from ‘ordinal values’

-          The so-called ‘mode’ is most probably the ‘median’

-          And I cannot really figure out what is the ‘std Dev’ when the input numbers are only on an ordinal scale type….

Verify throughout the text whevever the term ‘average’ is used that it is correctly used (ex. Page 7, line 208, on page 11, line 257 where the assertion ‘are less relevant’ should be ‘are perceived as less important’).

Figure 2 title is currently ‘Average importance of metrics by activity’: since this study is based on ‘Opinions-perceptions’ such a title should be ‘Perceived importance’ – similarly throughout this paper.

Page 11 – 2nd paragraph: the authors indicate that the most (perceived’ relevant metrics are ‘velocity, targeted value increase and work capacity’, and next that ‘velocity’ measures ‘the amount of work a team can deliver in a sprint’. As I have noted in a previous comment these three terms are the worst defined in Table 1, their basis mostly subjective with no right base measures. Furthermore, ‘velocity’ is classically a ratio of the distance by the time taken to cover that distance, measured in well defined and standardized measurement units (km and time in hours). However in Agile, ‘velocity’ the base measures in organizations are not the ones mentioned by the authors (‘amount of work’- not defined; and ‘during a Sprint – not standardized in terms of hours per Sprint) is most of the time the ‘actual hours’ divided by ‘estimated hours’: this give not a ‘velocity’ but a percentage of how good a team was at meeting an ‘estimate’  (and ‘estimation’  in Agile is very poor!) and it does not provide the ‘performance’ of a team as mentioned by the authors.

The above point out to a major weakness of this paper: yes, it presents well the ‘perceptions’ of practitioners, but there is no objective evaluation of the ‘metrics’ upon which opinions being surveyed.  The authors should clearly point out such a threat to the validity of such a such so that the readers are not led to adopt ‘widely perceived most relevant metrics’ as the right ones to use: some of these, based on a proper analysis could well be ‘placebos’ or worst, ‘snake-oil’ equivalent.

Author Response

We appreciate the review suggestions and comments received by the reviewer. These elements are key to improving the final quality of the manuscript. Below we respond to each issue raised.

Review #2

Title: The current title uses the term ‘Relevance’ (used 9 times in this paper’ while the Abstract refers to ‘Importance’ (used 25 times in this paper): therefore, the title should be modified to ‘importance’. Furthermore, this study input is the ‘opinion’ of the SCRUM practitioners’: therefore the title should be amended to ‘Perceived Importance’ instead of ‘Relevance’.

Author’s response: Thanks for your pertinent observation and suggestion. We agree that the best title for this paper is “Perceived Importance of Metrics for Agile Scrum Environments”.

Also verify the full text to ensure that the right terms (either relevance and importance) and used in a consistent manner throughout the text.

Author’s response: Thanks for your suggestion. We have used the word “importance” consistently in the paper.

Table 1: I was surprised to find out that this table does not include the metrics ‘Story Points’ widely used across the industry (and mentioned often as well in the literature), nor Function Points and COSMIC Functions Points, used by a number of practitioners and sometimes mentioned in the literature.

Author’s response: We appreciate the comment. We have used metrics such as “Estimated hours for a task” and “Remaining hours for a task”. Another alternative is to express them as “Estimated story points for a task” and “Remaining story points for a task”. Later they can be converted into hours, which is a unit of measurement that is especially useful for commercials in companies that interact with customers. In fact, it is virtually impossible to consider all the metrics, and all their derivations, addressed in the literature. Therefore, we have considered it as a limitation of this work and, consequently, we have exposed this situation in the limitations of the study and include it as a pertinent topic for future work.

Table 1: I was puzzled by the definitions of ‘velocity’ and ‘work capacity’ which use ‘amount of work’ and ‘sum of all work’ where ‘work’ has not been defined-measured anywhere (most previous definitions refer to ‘tasks’, ‘user stories’, etc.

Author’s response: We agree that it is very important to properly provide a clear definition of each metric. We have reviewed the definition of “velocity” and “work capacity” metrics. Velocity is defined as “Amount of user stories a team can tackle during a single sprint. It is calculated at the end of sprint by totaling the points for all fully completed user stories.” Work capacity is defined as “The total time the team is available for work during a sprint. It is usually measured in hours”. Capacity is how much availability the team has for the sprint. This may vary based on team members being on vacation, ill, etc. Capacity is based on the team's expected or projected future availability. Velocity is based on actual points completed, which is typically an average of all previous sprints. Velocity is used to plan how many product backlog items the team should bring into the next sprint.

Table 4: The authors did not explicitly described how the practitioners provided their opinions of the ‘relevance-importance’ of the ‘metrics’ listed: it appears however that they have used a range from 0 to 5, in increment of 1 (of course, this is my ‘guess’, and it may be something else…): to be noted that 0 to 5 in this context is an ‘ordinal scale’: each one being ‘more important’ than the previous ones, but these are ‘ordered labels’ and not ‘numerical values on the ratio-scale type where for instance 4 = 4 times greater than 1).  Therefore, if this is the range used by the authors in their questionnaire, than:

-           it is not allowed to calculate an ‘average’ from ‘ordinal values’

-          The so-called ‘mode’ is most probably the ‘median’

-          And I cannot really figure out what is the ‘std Dev’ when the input numbers are only on an ordinal scale type….

Author’s response: Thanks for your observation. We explained in the paper that the nominal scale was converted to a Likert ordinal scale from 1 to 5. Thus, it was possible to perform a descriptive statistical analysis considering the mean, mode, and standard deviation like it is presented in Table 4.

Verify throughout the text whevever the term ‘average’ is used that it is correctly used (ex. Page 7, line 208, on page 11, line 257 where the assertion ‘are less relevant’ should be ‘are perceived as less important’).

Author’s response: Thanks for your observation. We have followed your recommendation and changed it to “perceived as less important”.

Figure 2 title is currently ‘Average importance of metrics by activity’: since this study is based on ‘Opinions-perceptions’ such a title should be ‘Perceived importance’ – similarly throughout this paper.

Author’s response: Thanks for your suggestion. We have changed the title of Table 2 and performed the necessary changes throughout the manuscript.

Page 11 – 2nd paragraph: the authors indicate that the most (perceived’ relevant metrics are ‘velocity, targeted value increase and work capacity’, and next that ‘velocity’ measures ‘the amount of work a team can deliver in a sprint’. As I have noted in a previous comment these three terms are the worst defined in Table 1, their basis mostly subjective with no right base measures. Furthermore, ‘velocity’ is classically a ratio of the distance by the time taken to cover that distance, measured in well defined and standardized measurement units (km and time in hours). However in Agile, ‘velocity’ the base measures in organizations are not the ones mentioned by the authors (‘amount of work’- not defined; and ‘during a Sprint – not standardized in terms of hours per Sprint) is most of the time the ‘actual hours’ divided by ‘estimated hours’: this give not a ‘velocity’ but a percentage of how good a team was at meeting an ‘estimate’  (and ‘estimation’  in Agile is very poor!) and it does not provide the ‘performance’ of a team as mentioned by the authors.

Author’s response: Thanks for your suggestion. We have clarified the definitions of “velocity” and “work capacity”. These two new definitions clarify the differences between both metrics. Furthermore, we have reviewed the discussion section to better describe the relevance of these two metrics for Scrum. Two new references have been included:

Zuzek, T.; Kusar, J.; Rihar, L.; Berlec, T. Agile-Concurrent hybrid: A framework for concurrent product development using Scrum. Conc. Eng. 2020, 28, 255-264. https://doi.org/10.1177/1063293X20958541

Budacu, E.; Pocatilu, P. Real Time Agile Metrics for Measuring Team Performance. Inf. Econ. 2018, 22, 70-79. http://dx.doi.org/10.12948/issn14531305/22.4.2018.06

The above point out to a major weakness of this paper: yes, it presents well the ‘perceptions’ of practitioners, but there is no objective evaluation of the ‘metrics’ upon which opinions being surveyed.  The authors should clearly point out such a threat to the validity of such a such so that the readers are not led to adopt ‘widely perceived most relevant metrics’ as the right ones to use: some of these, based on a proper analysis could well be ‘placebos’ or worst, ‘snake-oil’ equivalent.

Author’s response: We appreciate the comments. We agree with the remarks, but we consider it is important to clarify the goals of this paper and the methodological limitations. Like any study that collects the opinion of users, the results obtained always represent the opinion of the sample and not of the entire population. This happens in any study that adopts the same approach. To mitigate this issue, it is important to have a relatively large and representative sample to reduce the risk of bias. This was the approach that the study took by including a sufficiently diverse sample considering the diverse profiles of the Scrum process stakeholders. Therefore, we have clarified the objectives of the study and the methodological limitations.

Reviewer 3 Report

This is a very interesting paper, appropriately structured and documented.

My suggestions to improve it are:

·        In Table 1: Overview of Scrum metrics, third column could receive more complete explanations  

·        For Table 4: Statistical analysis of the relevance of Scrum metrics, column Mode needs more precise explanation on Mode value calculation and its use.

·        Lines 245-246: Obtained results concerning H1, H2 & H3 hypothesis need more deep explanations to understand elaboration process and relation with your data.

·        The discussion section is very dense, it seems interesting to improve the layout, may be by introduction of bulleted lists, to show more clearly all interesting results and observations.

·        For different results I suggest to indicate corresponding data issued of your work and explain clear why you add references (may be proposing similar results).

·        In the lines 292-322more deep explanations related to your data could increase your trustfulness of your results.  

No comments

Author Response

We appreciate the review suggestions and comments received by the reviewer. These elements are key to improving the final quality of the manuscript. Below we respond to each issue raised.

Review #3

This is a very interesting paper, appropriately structured and documented.

Author’s response: Thank you very much for your positive feedback regarding this study.

My suggestions to improve it are:

  • In Table 1: Overview of Scrum metrics, third column could receive more complete explanations

Author’s response: Thanks for your suggestion. We have improved the definition of some Scrum metrics, particularly “Velocity” and “Work capacity” that were not properly defined. Velocity is the amount of user stories a team can tackle during a single sprint. It is calculated at the end of sprint by totaling the points for all fully completed user stories; while work capacity is the total time the team is available for work during a sprint. It is usually measured in hours. We have extended the definition of “Business Value” and “Number of impediments”.

For Table 4: Statistical analysis of the relevance of Scrum metrics, column Mode needs more precise explanation on Mode value calculation and its use.

Author’s response: Thanks for your observation. We have clarified how the mode was calculated. The mode represents the most frequent response given by the respondents.

Lines 245-246: Obtained results concerning H1, H2 & H3 hypothesis need more deep explanations to understand elaboration process and relation with your data.

Author’s response: Thanks for your suggestion. We have included a description regarding the significance of the test for each hypothesis.

The discussion section is very dense, it seems interesting to improve the layout, may be by introduction of bulleted lists, to show more clearly all interesting results and observations.

Author’s response: Thanks for your recommendation to reorganize the Discussion section. We have organized it to make clear the most relevant findings and its relevance for the evolution of theoretical and practical knowledge on the field. We have included two new subsections to better organized the discussion of the findings: (i) the importance of metrics; and (ii) the impact of control variables.

For different results I suggest to indicate corresponding data issued of your work and explain clear why you add references (may be proposing similar results). In the lines 292-322more deep explanations related to your data could increase your trustfulness of your results. 

Author’s response: Thanks for your suggestion. We have improved the Discussion section to better discuss the findings and associate data to demonstrate the relevance of the findings.

Round 2

Reviewer 2 Report

I have looked at the responses of the authors and to the modifications they made to their initial text. I have noticed a number of remaining major issues.

Issue 1:

The authors made the following modification: ‘A nominal scale (i.e., Very Low, Below Average, Average, Above Average, Very High) was considered and then converted for statistical analysis purposes to a Likert ordinal scale from 1 to 5.’

1: the sequence of terms ‘Very Low, Below Average, Average, Above Average, Very 194 High)’ is not a ‘nominal scale’: it is already an ‘ordinal’ scale with an obvious increase of ranking of each term’.

2. Indeed is it correct to convert it to a Likert scale for statistical analysis purposes, but it still remains on an ‘ordinal’ scale and only statistical operations with an ‘ordinal scale’ are then valid: such as calculation of the ‘median’. Therefore, my initial comment that it in invalid to use and report averages-mean (and standards deviations, etc.) such a done in this paper: the authors must correct such a deficiency throughout their paper (in both the text in Tables).

Issue 2:

The authors have made the following modification to their definition of ‘Velocity’: ‘Amount of user stories a team can tackle during a single sprint. It is calculated at the end of sprint by totaling the points for all fully completed user stories.’

1- I consider such a definition as highly confusing. In the first sentence, ‘amount of user stories’ corresponds to what? The term ‘amount’ would normally be interpreted as the ‘number’ or the ‘quantity’ of user stories. However, in the next sentence, the authors associate it to ‘total of points’ for all fully completed user stories, which term ‘points’ in ‘story points’ correspond to the ‘number of hours estimated’ at the beginning of an iteration or the beginning of a task, and not at completion time.

2- Furthermore, the authors in one of their response had indicated that they had avoided used ‘story points’ and had used ‘estimated hours instead. And still, they re-introduce ‘points’ in the definition of ‘velocity’.

3- The re-definition of Velocity  provided by the authors does not correspond to the most current definition of Velocity, that is ‘Story Points divided by actual hours, or ‘estimated hours’ divide by ‘actual hours’, which gives strictly a percentage % (which percentage – by definition - does not have a measurement unit).

4- In summary, the authors’ re-definition is incorrect.

5- Furthermore, this re-definition, as well as the other ones re-defined (number of impediments, business value, velocity, work capacity) while clarifying the text raise a serious methodological issue: while it adequately clarifies the text, these were not the definitions provided to the people surveyed: what were the definitions provided to the survey respondents, and discuss this in a validity threats section.

Issue 3:

The authors have increased the consistency within the text by using the term ‘importance’ rather than alternating between ‘importance’ and ‘relevance’: was the same inconsistency of terminology the same when asking survey respondents? If so, clarify and discuss in a a validity threats section.

Issue 4:

The authors have added the following in the Discussion section:Velocity is a widely known and used metric in Scrum that provides a vision regarding the amount of user stories a team can tackle during a single sprint [42-43].’

In practice, the authors provide here a different definition of ‘velocity’ than the one they used in Table 1, and it is even worst: ‘a vision’ is definitively not a sound quantity with repeatability, reproducibility, accuracy etc. Furthermore, the next part of the sentence that includes ‘can tackle’ is ill defined and not aligned with the definition in Table 1, and as I have mentioned above does not map to actual practice on calculating velocity as ‘estimated hours’ divided by ‘actual hours’.

And if the above definition of velocity come from references 42-43, these references should be deleted.

Issue 5:

The authors have added the following in the Discussion section: ‘Velocity provides a measure of the team's historical rate of delivering work. For example, it can be combined with other metrics such as backlog size, cycle time, or lead time, it becomes easier to predict future delivery dates or release milestones. This helps stakeholders and product owners to plan and set expectations more accurately. Another suggestion is to combine Velocity with metrics like Team Capacity and Individual Workload can help identify if the team is overburdened or underutilized. By analyzing the relationship between velocity and capacity, teams can make better decisions about how much work to take on in each sprint and allocate resources effectively.’

Considering that in practice ‘velocity’ in Agile is based on ‘story points’ or ‘estimated number of hours’ which is highly subjective, inconsistent across teams, across contexts, across almost anything, the end results of ‘velocity’ calculation cannot be comparable across any contexts: this is in quite contrast with the usage of ‘velocity’ in day-to-day life, where ‘velocity’ such as when driving, is strictly comparable across contexts – and controlled by laws, police, and calibrated measuring instruments.

Furthermore,  most users of ‘story points’ are poor ‘estimators’, and therefore the so-called ‘points’ or ‘estimated hours’ used in the calculation of ‘velocity’ in Agile  is a very unstable part of the calculation of ‘velocity’: the end result in ‘velocity’ is as weak as its weakest component.

In summary:

-          ‘velocity’ as calculated in practice in ‘agile’ is highly ‘unstable’ as well as ‘misleading’ : while it might ‘look’ and ‘shine’ like gold, it is not gold, but rather ‘fools gold’(e.g. pyrite) with none of the inherent properties of gold.’

-          Therefore, none of the inferences added above by the authors in the Discussion section rest on solid foundations and must be removed to avoid misleading the readers.

Author Response

We appreciate the updated review suggestions and comments received by the reviewer. These elements are very useful to improving the final quality of the manuscript. Below we respond to each issue raised.

Review #2

Issue 1:

The authors made the following modification: ‘A nominal scale (i.e., Very Low, Below Average, Average, Above Average, Very High) was considered and then converted for statistical analysis purposes to a Likert ordinal scale from 1 to 5.’

1: the sequence of terms ‘Very Low, Below Average, Average, Above Average, Very 194 High)’ is not a ‘nominal scale’: it is already an ‘ordinal’ scale with an obvious increase of ranking of each term’.

  1. Indeed is it correct to convert it to a Likert scale for statistical analysis purposes, but it still remains on an ‘ordinal’ scale and only statistical operations with an ‘ordinal scale’ are then valid: such as calculation of the ‘median’. Therefore, my initial comment that it in invalid to use and report averages-mean (and standards deviations, etc.) such a done in this paper: the authors must correct such a deficiency throughout their paper (in both the text in Tables).

Author’s response: Thanks for your clarification. We agree that it is not statistically correct to calculate the mean and standard deviation for an ordinal scale. However, in situations where the distance between ordinal elements is considered identical, it is possible to do and present the calculation. This has been a point of wide debate in the scientific community on platforms such as ResearchGate. Nevertheless, we also consider that it is more relevant to don’t present them, and in their alternative, to calculate the median. In this sense, Table 4 was revised and now presents the median and mode for each metric.

Issue 2:

The authors have made the following modification to their definition of ‘Velocity’: ‘Amount of user stories a team can tackle during a single sprint. It is calculated at the end of sprint by totaling the points for all fully completed user stories.’

1- I consider such a definition as highly confusing. In the first sentence, ‘amount of user stories’ corresponds to what? The term ‘amount’ would normally be interpreted as the ‘number’ or the ‘quantity’ of user stories. However, in the next sentence, the authors associate it to ‘total of points’ for all fully completed user stories, which term ‘points’ in ‘story points’ correspond to the ‘number of hours estimated’ at the beginning of an iteration or the beginning of a task, and not at completion time.

2- Furthermore, the authors in one of their response had indicated that they had avoided used ‘story points’ and had used ‘estimated hours instead. And still, they re-introduce ‘points’ in the definition of ‘velocity’.

3- The re-definition of Velocity provided by the authors does not correspond to the most current definition of Velocity, that is ‘Story Points divided by actual hours, or ‘estimated hours’ divide by ‘actual hours’, which gives strictly a percentage % (which percentage – by definition - does not have a measurement unit).

4- In summary, the authors’ re-definition is incorrect.

5- Furthermore, this re-definition, as well as the other ones re-defined (number of impediments, business value, velocity, work capacity) while clarifying the text raise a serious methodological issue: while it adequately clarifies the text, these were not the definitions provided to the people surveyed: what were the definitions provided to the survey respondents, and discuss this in a validity threats section.

Author’s response: We agree that the definition of the "Velocity" metric should be clarified. It is important to emphasize that both proposals made by the reviewer are correct. It can be calculated considering the Story Points divided by actual hours or looking to the estimated hours divided by actual hours. In our first definition we have used the general term “amount” that can be applied to both situations. Companies tend to adopt one approach or another. To clarify the definition of “Velocity” we have adopted the same definition as provided by Agile Aacdemy (https://www.agile-academy.com/en/scrum-master/velocity-definition-and-how-you-can-calculate-it/). We have also provided more information how Scrum metrics can be calculated. Furthermore, the changes made to the definition of the metrics arose from the suggestions of the reviewers with the intention of clarifying and giving some more information about their calculation process. There was no change in their original meaning. Additionally, we included in the Conclusions section the indication of an important limitation that is related to the definition of metrics. We found throughout this study that there is not total agreement on the definition of metrics and their way of implementation in organizations. In this sense, the perception of the importance of metrics is strongly related to the way these metrics were implemented in the respondents' organizations and not in the formal definition associated with each one of them.

Issue 3:

The authors have increased the consistency within the text by using the term ‘importance’ rather than alternating between ‘importance’ and ‘relevance’: was the same inconsistency of terminology the same when asking survey respondents? If so, clarify and discuss in a a validity threats section.

Author’s response: In the previous revision process we followed the reviewer's recommendation and made this change because we also felt that the term "importance" might be better suited to the context of the study. However, the interpretation of both expressions is equivalent. This vision is also confirmed in dictionaries like Thesaurus, MacMillan, and Merriam-Webster.

https://www.thesaurus.com/browse/relevance

https://www.macmillandictionary.com/thesaurus-category/british/relevant-and-appropriate

https://www.merriam-webster.com/thesaurus/relevance

Issue 4:

The authors have added the following in the Discussion section: ‘Velocity is a widely known and used metric in Scrum that provides a vision regarding the amount of user stories a team can tackle during a single sprint [42-43].’

In practice, the authors provide here a different definition of ‘velocity’ than the one they used in Table 1, and it is even worst: ‘a vision’ is definitively not a sound quantity with repeatability, reproducibility, accuracy etc. Furthermore, the next part of the sentence that includes ‘can tackle’ is ill defined and not aligned with the definition in Table 1, and as I have mentioned above does not map to actual practice on calculating velocity as ‘estimated hours’ divided by ‘actual hours’.

And if the above definition of velocity come from references 42-43, these references should be deleted.

Author’s response: We have decided to eliminate reference to these two studies to avoid confusion in interpreting the previously established definition of " Velocity". Instead, we have stated that Velocity is a widely known and used metric in Scrum that measures the work rate. This vision is supported in the following reference:

Bansai, S. Velocity – An Agile Metrics. Available online: https://www.izenbridge.com/blog/velocity-in-agile-scrum

Issue 5:

The authors have added the following in the Discussion section: ‘Velocity provides a measure of the team's historical rate of delivering work. For example, it can be combined with other metrics such as backlog size, cycle time, or lead time, it becomes easier to predict future delivery dates or release milestones. This helps stakeholders and product owners to plan and set expectations more accurately. Another suggestion is to combine Velocity with metrics like Team Capacity and Individual Workload can help identify if the team is overburdened or underutilized. By analyzing the relationship between velocity and capacity, teams can make better decisions about how much work to take on in each sprint and allocate resources effectively.’

Considering that in practice ‘velocity’ in Agile is based on ‘story points’ or ‘estimated number of hours’ which is highly subjective, inconsistent across teams, across contexts, across almost anything, the end results of ‘velocity’ calculation cannot be comparable across any contexts: this is in quite contrast with the usage of ‘velocity’ in day-to-day life, where ‘velocity’ such as when driving, is strictly comparable across contexts – and controlled by laws, police, and calibrated measuring instruments.

Furthermore,  most users of ‘story points’ are poor ‘estimators’, and therefore the so-called ‘points’ or ‘estimated hours’ used in the calculation of ‘velocity’ in Agile  is a very unstable part of the calculation of ‘velocity’: the end result in ‘velocity’ is as weak as its weakest component.

In summary:

-          ‘velocity’ as calculated in practice in ‘agile’ is highly ‘unstable’ as well as ‘misleading’ : while it might ‘look’ and ‘shine’ like gold, it is not gold, but rather ‘fools gold’(e.g. pyrite) with none of the inherent properties of gold.’

-          Therefore, none of the inferences added above by the authors in the Discussion section rest on solid foundations and must be removed to avoid misleading the readers.

Author’s response: We agree with the reviewer's view, and this was precisely the idea we had in writing this text. Perhaps the way it was written was not clear and led to some confusion. The idea that we wanted to convey is that "Velocity" is an inaccurate metric and should not be used in isolation. It is indeed a metric that is widely used because it is easy to calculate, but it is not advisable to use it alone. We have clarified it in the discussion section.